# TAVGBench: Benchmarking Text to Audible-Video Generation

## ABSTRACT

The Text to Audible-Video Generation (TAVG) task involves generating videos with accompanying audio based on text descriptions. Achieving this requires skillful alignment of both audio and video elements. To support research in this field, we have developed a comprehensive Text to Audible-Video Generation Benchmark (TAVGBench), which contains over 1.7 million clips with a total duration of 11.8 thousand hours. We propose an automatic annotation pipeline to ensure each audible video has detailed descriptions for both its audio and video contents. We also introduce the Audio-Visual Harmoni score (AVHScore) to provide a quantitative measure of the alignment between the generated audio and video modalities. Additionally, we present a baseline model for TAVG called TAVDiffusion, which uses a two-stream latent diffusion model to provide a fundamental starting point for further research in this area. We achieve the alignment of audio and video by employing cross-attention and contrastive learning. Through extensive experiments and evaluations on TAVGBench, we demonstrate the effectiveness of our proposed model under both conventional metrics and our proposed metrics.

## CCS CONCEPTS

• **Computing methodologies** → **Image and video acquisition**.

## KEYWORDS

Text to Audible-Video Generation Benchmark (TAVGBench), Text to Audible-Video Diffusion (TAVDiffusion)

## 1 INTRODUCTION

The text to video generation task [2, 10, 13, 39, 47, 51] has been boosted through the integration of computer vision and natural language processing. This task translates textual descriptions into visual representations, enriching multimedia experiences, and improving accessibility for individuals with visual impairments. However, we observe that although existing methods excel in converting textual descriptions into visual content, the endeavor to integrate synchronized audio into these videos remains largely unexplored. This gap underscores a fundamental necessity within the realm of multimodal generation—the imperative to generate video content with auditory components guided solely by textual descriptions.

In this paper, considering the clear gap in current research, we introduce a new task: Text to Audible-Video Generation (TAVG). This task marks a significant change, requiring models to move beyond

*ACM MM, 2024, Melbourne, Australia*

© 2024 Copyright held by the owner/author(s). Publication rights licensed to ACM.
ACM ISBN 978-x-xxxx-xxxx-x/YY/MM
https://doi.org/10.1145/nnnnnnn.nnnnnnn

just generating visual content and also creating audio alongside it. Unlike typical text to video tasks that only focus on unimodal video generation, TAVG requires generating both audio and video at the same time, guided by written descriptions. By taking on this task, we aim to push the boundaries of multimodal generation, making it possible to create immersive audio-visual experiences using only text prompts. The task definition is shown in Fig. 1.

To successfully achieve TAVG, a comprehensive dataset with well-aligned audio and video components is essential. However, we find no mature benchmark available for this task to support training and testing, primarily attributable to the absence of such a large-scale dataset. Building upon the foundation of TAVG, we propose establishing a Text to Audible-Video Generation Benchmark (TAVGBench), which allows the model to be trained in a supervised manner. At the core of TAVGBench lies a carefully selected dataset, comprising diverse textual descriptions and their corresponding audio-visual pairs. This dataset facilitates comprehensive evaluation and comparison of various methods. Our dataset consists of over 1.7 million audio-visual pairs sourced from YouTube videos. We design a coarse-to-fine pipeline to automatically achieve text annotation for audio-visual pairs in the dataset. Specifically, we utilize BLIP2 [23] and WavCaps [27] to describe the video and audio components, respectively. Additionally, we employ ChatGPT [29] to rephrase and integrate annotations from both modalities, which enables our annotation pipeline to excel in understanding context and producing human-like text descriptions. To evaluate the alignment degree between the generated audio and video, we introduce a new metric for the TAVG task to measure the harmony of the generated results, called the Audio-Visual Harmoni score (AVHScore). This metric quantifies the alignment between video and audio in a multi-modal, high-dimensional semantic space.

Utilizing our proposed TAVGBench, we present a Text to Audible-Video Diffusion (TAVDiffusion) model as the baseline method. This method is based on the latent diffusion model [34], representing an initial attempt to generate audio and video from text. Given the requirement of multimodal alignment, we propose two strategies to achieve the alignment of multimodal latent variables from the perspective of feature interaction and feature constraints. We extensively evaluate the baseline model against the proposed TAVGBench using both the conventional metrics and our proposed metrics and demonstrate the effectiveness of our method in the task of TAVG.

As follows, we summarize our main contributions as:

- We introduce the TAVG task, extending multimodal generation by integrating synchronized audio with visual content, addressing a crucial research gap.
- We present TAVGBench, a large-scale benchmark dataset with an automatic text description annotation pipeline and a novel Audio-Visual Harmoni score (AVHScore), significantly facilitating the TAVG task.
- We propose the Text to Audible-Video Diffusion (TAVDiffusion) model as a baseline method built upon the latent diffusion model.

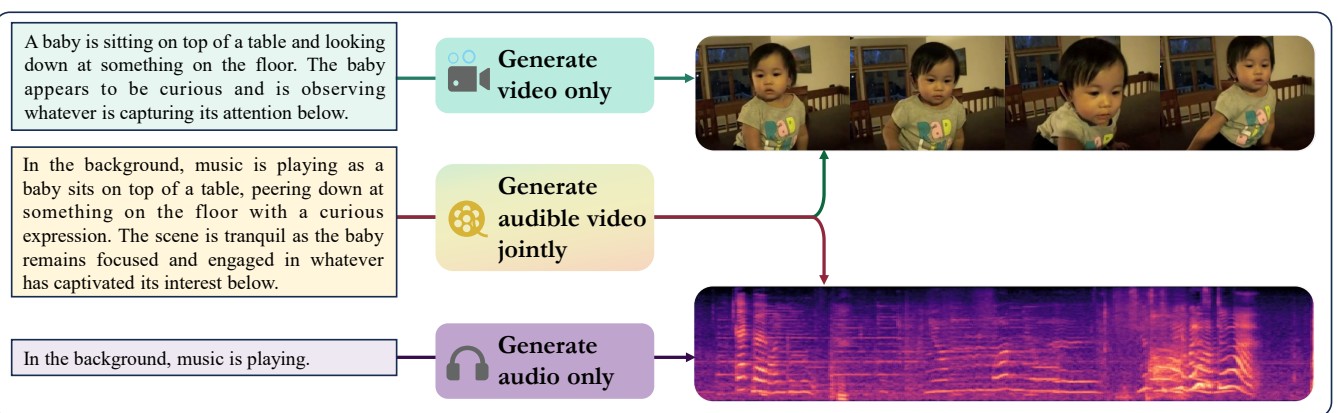

**Figure 1: Comparison of the proposed TAVG task with existing generation tasks. (a) Text to video generation (TVG) generates the corresponding videos through text descriptions. (b) Text to audio generation (TAG) generates the corresponding audio through text descriptions. (c) The proposed TAVG task generates audio-visual content based on text descriptions of both audio and video elements.**

## 2 RELATED WORK

**Text to video generation.** Text to video generation [2, 10, 13, 18, 39, 47, 49, 51] presents a challenging and extensively studied task. Previous studies utilize various generative models, such as GANs [24], and Autoregressive models [15, 55]. In recent years, the emergence of diffusion models [14] in content generation (*i.e.* text-to-image generation) catalyzes substantial advancements in text-to-video generation research. Imagen-Video [13], Make-A-Video [39], and show-1 [56] propose a deep cascade of temporal and spatial up-samplers for video generation, training their models jointly on both image and video datasets. The majority of subsequent works are grounded in the latent diffusion model [34], leveraging pre-trained UNet weights on 2D images. VideoLDM [2] adopts a latent diffusion model, where a pre-trained latent image generator and decoder are fine-tuned to ensure temporal coherence in generated videos. LAVIE [47] integrates Rotary Positional Encoding (RoPE) [42] into the network to capture temporal relationships among video frames. AnimateDiff [10] employs a strategy of freezing a pre-trained latent image generator while exclusively training a newly inserted motion modeling module. SimDA [51] proposes an efficient temporal adapter to help a trained 2D diffusion model extract temporal information. These advancements lay the foundation for an efficient multimodal diffusion pipeline.

**Text to audio generation.** Similar to video generation, the task of text to audio generation also evolves from GANs [28, 37] and Autoregressive models [20] to diffusion models. DiffSound [53] proposes a VQVAE model and a mask-based text generation strategy to address scarce audio-text paired data, albeit with potentially limited performance due to the lack of detailed text information. AudioGen [20] employs an autoregressive framework utilizing a Transformer-based decoder to generate tokens directly from the waveform. It applies data augmentation and simplifies language descriptions into labels, sacrificing detailed temporal and spatial information. AudioLDM [25] transfers the latent diffusion model from the domain of visual generation to text-to-audio generation. It

encodes text information through CLAP embedding [6] to achieve guidance. Tango [8] follows the LDM pipeline and replaces the CLAP to T5 [33] for more expressive text embedding.

**Audio video mutual/joint generation.** In addition to text-guided content generation, the mutual or joint generation of audio and video gradually become the focus of research in recent years. Typically, audio and video modalities serve as conditional signals for each other to achieve mutual generation, namely generating audio from video or video from audio. For the former, SpecVQGAN [16], CondFoleyGen [5], and Diff-Foley [26] implement audio generation from video using VQGAN, autoregressive transformer, and diffusion model, respectively. Regarding the latter, soundini [22] utilizes audio as a control signal to guide the video diffusion model for video editing purposes. Sung *et al.* [43] constrain the generated video content from audio to align more closely with the original audio using contrastive learning. TempoTokens [54] introduces an AudioMapper that employs a token encoded by a pre-trained audio encoder as a condition for enabling audio-to-video generation within a diffusion framework.

Based on the mutual generation of the two modalities, several studies explore the joint generation of audible video content. MM-diffusion [36] employs a diffusion UNet that takes inputs and outputs from both modalities, enabling the joint generation of two modalities for the first time. Zhu *et al.* [58] employ a video diffusion architecture to generate video and then retrieve audio, presenting an alternative approach to joint generation. Xing *et al.* [50] propose augmenting the existing diffusion model with optimization operations during the inference process to achieve audio-video generation while maintaining alignment.

**Uniqueness of our benchmark.** Despite the extensive exploration of multimodal generation tasks, there currently lacks a comprehensive benchmark and large-scale dataset specifically for the text to audible-video generation task. Addressing this gap, our solution offers a dataset for both training and evaluation, alongside metrics for assessing multimodal alignment. Additionally, we provide a straightforward baseline method.

**Table 1: Statistics of TAVGBench and other existing video generation datasets. The terms "audio" and "video" represent the modality described by the description in the dataset. The terms "Sentence" and "Word" represent the average number of sentences and words per video annotation, respectively.**

| Dataset | Sample | | Description | | | | |
|---|---|---|---|---|---|---|---|
| | Clip | DUR. (h) | Audio | Video | Method | Sentence | Word |
| AudioCaps [19] | 46K | 5.3 | ✔ | ✘ | Human-written | 1.0 | 9.03 |
| MSR-VTT [52] | 10K | 41.2 | ✘ | ✔ | Human-written | 20.0 | 185.7 |
| WebVid [1] | 10M | 53K | ✘ | ✔ | Automatic caption | 1.0 | 12.0 |
| FAVDBench [38] | 11.4K | 24.4 | ✔ | ✔ | Human-written | 12.6 | 218.9 |
| TAVGBench | 1.7M | 11.8K | ✔ | ✔ | Automatic caption | 2.32 | 49.98 |

## 3 THE TAVGBENCH

### 3.1 Dataset statistics

The TAVG task entails the generation of audible videos guided by input text prompts. To support this task, we introduce a benchmark named TAVGBench. Our dataset is sourced from AudioSet [7], comprising 2 million aligned audio-video pairs obtained from YouTube. After excluding invalid videos, we obtained 1.7 million pieces of original data. Each video sample has a duration of 10 seconds, contributing to a total video duration of 11.8K hours in the dataset. To provide a comprehensive understanding of the scale and characteristics of our dataset, we compare it with the datasets from other related tasks. Table 1 presents a comparative analysis of TAVG-Bench against these datasets in terms of size, source, and other relevant attributes.

It can be observed from Table 1 that the AudioCaps [19], MSR-VTT [52], and WebVid [1] only describe the content of a single modality (only audio or video modality). Although FAVDBench describes two modalities, the scale of the dataset is limited. The TAVGBench that we propose takes into account descriptions of both audio and video modalities while ensuring a sufficiently large dataset scale. In addition, the videos in WebVid have watermarks, which greatly restrict their application in actual scenarios. This comparison highlights the scale and unique characteristics of the TAVG-Bench dataset, emphasizing its potential for advancing research in audible-video generation. Additionally, TAVGBench exhibits a balanced distribution of textual descriptions, with an average of 2.32 sentences and 49.98 words per video annotation, providing substantial contextual information for each clip. These comparative statistics underscore TAVGBench's extensive scale, multimodal nature, and linguistic richness, positioning it as a valuable resource for advancing research in our TAVG task.

### 3.2 Annotation details

Given the absence of detailed text annotations in AudioSet [7] for both its video and audio content, we implement a coarse-to-fine pipeline to automate the generation of text descriptions. The complete pipeline is shown in Fig. 2. Initially, we employ two sophisticated methods, namely BLIP2 [23] for video description and WavCaps [27] for audio description, to annotate the video and audio components, respectively.

However, despite the effectiveness of these methods in capturing the essence of video and audio content, the generated annotations often lacked coherence and context. To address this limitation and improve the overall quality of the annotations, we introduced a refinement step using ChatGPT [29], a powerful language model capable of paraphrasing and enriching textual input.

During the refinement phase, we utilize ChatGPT to rephrase and enhance the annotations generated by BLIP2 and WavCaps. By feeding the initial annotations into the ChatGPT model, we obtain revised annotations that exhibit enhanced coherence, contextual relevance, and linguistic refinement. Initially, we individually rephrase the video and audio descriptions to rectify grammatical errors and enhance descriptive content. Subsequently, we employ ChatGPT to amalgamate the descriptions from both modalities into a unified, coherent sentence. This iterative process not only enhances the readability of the annotations but also ensures consistency and accuracy throughout the entire annotation corpus.

Incorporating ChatGPT into our pipeline has significantly enhanced the ability to detect subtle nuances and semantic complexities within the video and audio content. As a result, our annotation pipeline excels in understanding context and producing human-like text descriptions, thus facilitating the creation of annotations that more precisely capture the essence of the underlying content.

### 3.3 Evaluation metric

Existing metrics for video (FVD, KVD [45]) and audio (FAD [36]) generation primarily focus on the quality of each modality separately. However, for The TAVG task, we not only need to generate high-quality audio and video but also need to ensure that these two modalities are accurately synchronized. To address the necessity for evaluating the alignment degree between generated audio and video, we propose a novel metric called the Audio-Visual Harmony Score (AVHScore). This metric quantifies the alignment of audio-video pairs by calculating the product of the extracted audio-visual features. We use a robust feature extractor (ImageBind [9]) to project the video frames and the audio into a unified feature space. Formally, the we define the AVHScore $S_{\text{AVH}}$ as follows:

$$S_{\text{AVH}} = \frac{1}{n} \sum_{i=1}^{N} \cos(E_v(V_i), E_a(A)) \quad (1)$$

where cos denotes cosine similarity. $E_v$ and $E_a$ represent the vision encoder and audio encoder, respectively, in the ImageBind model. $N$ signifies the number of video frames, and we compute the similarity between each video frame and the corresponding audio input, averaging the results across all frames.

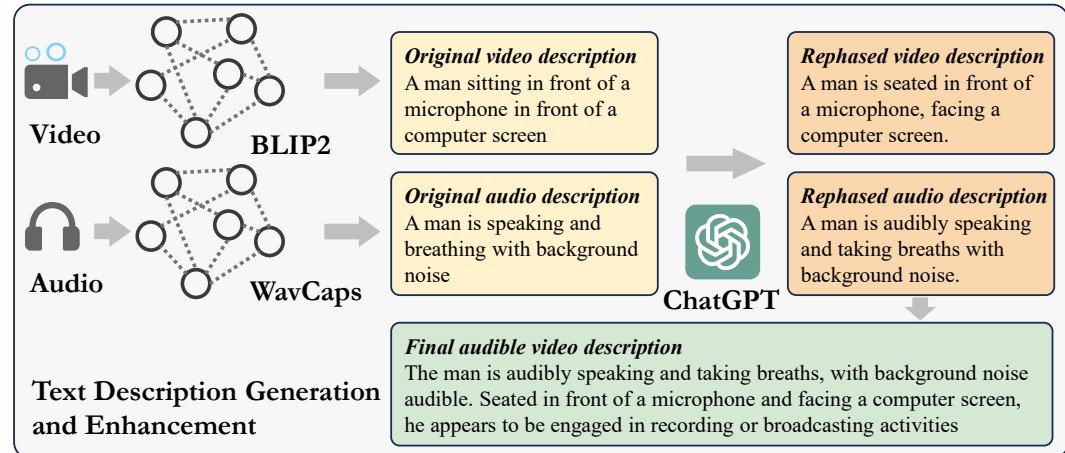

**Figure 2: Overview of the annotation pipeline. We employ BLIP2 for video descriptions and WavCaps for audio descriptions. The descriptions are further refined using ChatGPT, resulting in the final detailed audible video description.**

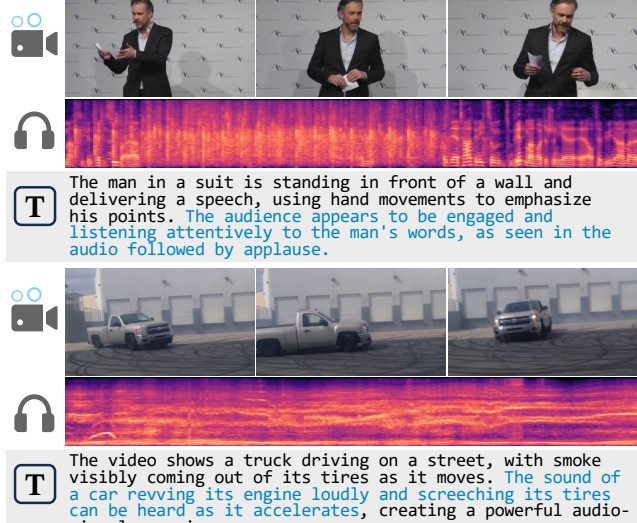

The man in a suit is standing in front of a wall and delivering a speech, using hand movements to emphasize his points. The audience appears to be engaged and listening attentively to the man's words, as seen in the audio followed by applause.

The video shows a truck driving on a street, with smoke visibly coming out of its tires as it moves. The sound of a car revving its engine loudly and screeching its tires can be heard as it accelerates, creating a powerful audio-visual experience.

**Figure 3: Data samples. The video (we give three frames for each video clip), audio, and the corresponding generated captions. We highlight the video caption in black and the audio caption in blue.**

## 4  A BASELINE

We propose a new baseline method for the text to audible-video generation (TAVG) task as shown in Fig. 4, named TAVDiffusion. The entire network structure is based on the latent diffusion model [34].

### 4.1  Preliminary: Latent diffusion model

The latent diffusion model follows the standard formulation outlined in DDPM [14], which comprises a forward diffusion process and a backward reverse denoising process. Initially, a data sample $\mathbf{x} \sim p(\mathbf{x})$ undergoes processing by an autoencoder, consisting of an encoder $\mathcal{E}$ and a decoder $\mathcal{D}$. The autoencoder projects $\mathbf{x}$ into a latent variable $\mathbf{z}$ via $\mathbf{z} = \mathcal{E}(\mathbf{x})$. Subsequently, the diffusion and

denoising process takes place within the latent space. The denoised latent variable is recovered to the input space by $\hat{\mathbf{x}} = \mathcal{D}(\hat{\mathbf{z}}_0)$.

Inspired by non-equilibrium thermodynamics, diffusion models [14, 40] are a class of latent variable $(z_1, ..., z_T)$ models of the form $p_\theta(z_0) = \int p_\theta(z_{0:T})dz_{1:T}$, where the latent variables are of the same dimensionality as the input data $z_0$. The joint distribution $p_\theta(z_{0:T})$ is also called the *reverse process*:

$$p_\theta(z_{0:T}) = p_\theta(z_T) \prod_{t=1}^{T} p_\theta(z_{t-1}|z_t), \quad (2)$$

where

$$p_\theta(z_{t-1}|z_t) = \mathcal{N}(z_{t-1}; \mu_\theta(z_t, t), \Sigma_\theta(z_t, t)). \quad (3)$$

Here, $\mu_\theta$ and $\Sigma_\theta$ are determined through a denoiser network $\epsilon_\theta(z_t, t)$, typically structured as a UNet [35].

The approximate posterior $q(z_{1:T}|z_0)$ is called the *forward process*, which is fixed to a Markov chain that gradually adds noise according to a predefined noise scheduler $\beta_{1:T}$:

$$q(z_{1:T}|z_0) = \prod_{t=1}^{T} q(z_t|z_{t-1}), \quad (4)$$

where

$$q(z_t|z_{t-1}) = \mathcal{N}(z_t; \sqrt{1 - \beta_t}z_{t-1}, \beta_t \mathbf{I}). \quad (5)$$

The training is performed by minimizing a variational bound on negative log-likelihood:

$$\begin{aligned}
\mathbb{E}_q[-\log p_\theta(z_0)] &\leq \mathbb{E}_q[-\log \frac{p_\theta(z_{0:T})}{q(z_{1:T}|z_0)}] \\
&= \mathbb{E}_q[D_{KL}(q(z_T|z_0)\|p(z_T)) \\
&\quad + \sum_{t>1} D_{KL}(q(z_{t-1}|z_t, z_0)\|p_\theta(z_{t-1}|z_t)) \\
&\quad - \log p_\theta(z_0|z_1)].
\end{aligned} \quad (6)$$

Hence, the final training objective of $\theta$ is a noise estimation loss, with a conditional variable $\mathbf{c}$, it can be formulated as:

$$\mathcal{L}_{\text{diffusion}}(\theta) := \mathbb{E}_{\mathbf{z},\mathbf{c},\epsilon \sim \mathcal{N}(0,\mathbf{I}),t} \left[ \|\epsilon - \epsilon_\theta(\mathbf{z}_t, \mathbf{c}, t)\|^2 \right]. \quad (7)$$

**Figure 4: Overview of the TAVDiffusion training (left) and inference (right) stages. We develop a two-stream architecture. During the training phase, we randomly select a timestep $t$ and employ diffusion loss to guide the single-step denoising. In the inference phase, iterative denoising is conducted to finally produce an audible video.**

## 4.2 TAVDiffusion

With the forward and reverse process in the latent diffusion model defined in Sec. 4.1, we further present the baseline two-stream diffusion pipeline for joint text to audible-video diffusion.

**Multimodal Latent encoders.** We employ two independent latent autoencoders for our multimodal input to conduct latent space encoding and decoding. This process can be formulated as:

$$\text{Encoder:} \quad \mathbf{z}_a = \mathcal{E}_a(\mathbf{x}_a), \mathbf{z}_v = \mathcal{E}_v(\mathbf{x}_v),$$
$$\text{Decoder:} \quad \hat{\mathbf{x}}_a = \mathcal{D}_a(\hat{\mathbf{z}}_{a,0}), \hat{\mathbf{x}}_v = \mathcal{D}_v(\hat{\mathbf{z}}_{v,0}), \tag{8}$$

where the subscripts $a$ and $v$ represent audio and video modality, respectively. Specifically, for $D_v$, we utilize the pre-trained autoencoder in stable diffusion [34] along with its weights. Given an original video with dimensions $T \times 3 \times H_v \times W_v$, the size of the latent variable becomes $T \times 4 \times \frac{H_v}{8} \times \frac{W_v}{8}$. As for $D_a$, we employ the pre-trained autoencoder from AudioLDM [25] to encode the audio mel spectrogram into the latent space. For the dimensions $1 \times H_a \times W_a$ of the input mel spectrogram, the size of the latent variable becomes $8 \times \frac{H_a}{8} \times \frac{W_a}{8}$.

**Multimodal diffusion process.** For the input of audio and video modalities, we use a two-stream structure to perform the forward and reverse diffusion process for the latent variables $\mathbf{z}_a, \mathbf{z}_v$, as shown in Fig. 4. Unlike vanilla diffusion where a single modality is generated, we aim to simultaneously recover two consistent modalities (i.e. audio and video) within a single diffusion process.

We consider that the reverse and forward processes of each modality are independent because they have distinct distributions. Taking the audio latent variable $\mathbf{z}_a$ as an illustration, its reverse process at timestep $t$ is defined as:

$$p_{\theta_a}(z_{a,t-1}|z_t) = \mathcal{N}(z_{a,t-1}; \mu_{\theta_a}(z_{a,t}, t), \Sigma_{\theta_a}(z_{a,t}, t)). \tag{9}$$

The forward process at timestep $t$ is defined as follows:

$$q(z_{a,t}|z_{a,t-1}) = \mathcal{N}(z_{a,t}; \sqrt{1-\beta_t}z_{t-1}, \beta_t\mathbf{I}). \tag{10}$$

For brevity, we omit the reverse and forward process for video $\mathbf{z}_v$ via $\theta_v$, as it shares a similar formulation. It is important to note that we empirically set a shared schedule for hyper-parameters $\beta$ across audio and video to streamline the process definition.

Summarizing the formulations above, the final definition of the multimodal diffusion loss is:

$$\mathcal{L}_{\text{diffusion}}(\theta_a, \theta_v) = \mathcal{L}_{\text{diffusion}_a}(\theta_a) + \mathcal{L}_{\text{diffusion}_v}(\theta_v)$$
$$:= \mathbb{E}_{\mathbf{z}_a, \mathbf{c}, \epsilon_a \sim \mathcal{N}(0, \mathbf{I}), t}\left[\left\|\epsilon_a - \epsilon_{\theta_a}\left(\mathbf{z}_{a,t}, \mathbf{c}, t\right)\right\|^2\right] \tag{11}$$
$$+ \mathbb{E}_{\mathbf{z}_v, \mathbf{c}, \epsilon_v \sim \mathcal{N}(0, \mathbf{I}), t}\left[\left\|\epsilon_v - \epsilon_{\theta_v}\left(\mathbf{z}_{v,t}, \mathbf{c}, t\right)\right\|^2\right].$$

In our task, the conditional variable $\mathbf{c}$ denotes the text embedding of the input, and we use the CLIP [32] text encoder with its tokenizer to obtain the text embedding.

**Diffusion UNet architecture.** For $\theta_a$ and $\theta_v$, we employ two parallel UNet branches to perform denoising for audio and video latent variables, respectively. For the audio UNet $\theta_a$, we directly utilize the UNet architecture from stable diffusion [34] and adjust the number of channels in its input layer from 4 to 8. Regarding the video UNet $\theta_v$, akin to AnimateDiff [10], we adapt the 2D convolution in the 2D UNet of stable diffusion[1] to a pseudo-3D convolution, which comprises a 1D convolution followed by a 2D convolution. Additionally, we incorporate the Temporal Transformer layer [46] into the network. These changes allow the network to accept video as input and keep its temporal details.

## 4.3 Multimodal interaction

In this section, we introduce a methodology for aligning the intermediate features $f_a, f_v$, associated with $\theta_a, \theta_v$ respectively, by

---

[1]We use stable diffusion version 1.5 and its pre-trained weights as partial initialization.

employing a cross-attention mechanism [46]. Our approach aims to enhance the performance of joint generation tasks involving video and audio by synchronizing the information flow between the two modalities during the diffusion process. Given $f_a$ and $f_v$, we represent the process of cross-attention as follows and utilize the resulting $\hat{f}_a$ and $\hat{f}_v$ as the input of the diffusion UNet decoder.

$$\hat{f}_a = \text{CrossAtten}(f_a, f_v), \ \hat{f}_v = \text{CrossAtten}(f_v, f_a). \quad (12)$$

## 4.4 Multimodal alignment

The feature interaction mechanism does not explicitly enforce the alignment of multimodal features. Hence, it is crucial to integrate a loss function that guarantees alignment between the feature representations of audio and visual modalities. To tackle this issue, we propose an Explicit audio-visual Alignment Strategy (EAS) based on contrastive learning [11].

In a multimodal scenario, where we have feature vectors $(\hat{f}_a, \hat{f}_v)$ representing two modalities, contrastive learning is employed to model the alignment between these modalities. The underlying assumption is that features that align well with each other should be brought closer together, whereas those that contradict each other should be pushed apart. We define alignment as a bidirectional process, leading to the formulation of the contrastive loss as follows:

$$\mathcal{L}_{\text{EAS}}(\hat{f}_a, \hat{f}_v) = -\log \frac{\exp(s(\hat{f}_a, \hat{f}_a^p)/\tau)}{\sum_{\hat{f}_v} \exp(s(\hat{f}_a, \hat{f}_v)/\tau)} \\ -\log \frac{\exp(s(\hat{f}_v, \hat{f}_v^p)/\tau)}{\sum_{\hat{f}_a} \exp(s(\hat{f}_v, \hat{f}_a)/\tau)}, \quad (13)$$

where $\tau = 0.1$ is the temperature scalar, $\hat{f}_a^p$ ($\hat{f}_v^p$) is the positive sample corresponding to $\hat{f}_a$ ($\hat{f}_v$) from the other modality. The function $s(\cdot, \cdot)$ computes the cosine similarity. We generate positive and negative samples based on the feature representations of two modalities. Specifically, we define embeddings of the two modalities with the same index (paired data) as positive samples, while those with different indexes are considered negative samples (unpaired data).

The bottleneck of contrastive learning lies in designing the positive/negative pairs with effective similarity measure, *i.e.* $s(\cdot, \cdot)$ in our case. We use a linear projection with softmax activation $l_\beta(\cdot)$ to calculate similarity weight based on the input of a specific modality [4] different information contained in different tokens. Given two modalities $(a, v)$, a weighted similarity function $s(\cdot, \cdot)$ is:

$$s(\hat{f}_a, \hat{f}_v) = \sum_{\hat{f}_a} l_{\beta_a}(\hat{f}_a)\cos(\hat{f}_a, \hat{f}_v) + \sum_{\hat{f}_v} l_{\beta_v}(\hat{f}_v)\cos(\hat{f}_v, \hat{f}_a). \quad (14)$$

## 4.5 Objective Function

The objective functions are divided into task loss ($\mathcal{L}_{\text{diffusion}}$) and feature alignment loss ($\mathcal{L}_{\text{EAS}}$). The total loss is a weighted sum of the above terms:

$$\mathcal{L} = \mathcal{L}_{\text{diffusion}} + \lambda \mathcal{L}_{\text{EAS}}, \quad (15)$$

where $\lambda$ represents the balanced weights during training. Empirically, the loss weight is set as $\lambda = 0.1$.

# 5 EXPERIMENTAL RESULTS

## 5.1 Implementation details

**Datasets.** We train our model on the TAVGBench dataset that we introduced. For the evaluation phase, we select 3,000 samples from the TAVGBench evaluation subset. Additionally, we evaluate the performance of our model on the test subset of FAVDBench [38], which comprises 1,000 samples. The FAVDBench provides more fine-grained audible video descriptions, enabling the generation of more detailed videos. Importantly, because the data of FAVDBench is not utilized in the training phase, we could assess the *zero-shot* capabilities of our model based on its performance on FAVDBench.

**Training Details.** We implement TAVDiffusion using PyTorch [30]. We adopt a video training resolution of $256 \times 320 \times 20$ to balance training efficiency and motion quality. We convert the audio into a mel spectrogram and perform training and inference on it. We use the MelGAN Vocoder [21] to convert the denoised audio mel spectrogram into the audio waveform. We use a learning rate of $1 \times 10^{-4}$ and train the model with 64 NVIDIA A100 GPUs for $1.0 \times 10^5$ steps. At inference time, we apply the DDIM scheduler [41] and only sample 50 timesteps.

**Evaluation Metrics.** We begin by separately measuring the quality of the generated audio and video. To evaluate the video, we adopt the Frechet Video Distance (FVD) [45], Kernel Video Distance (KVD) [44], and CLIPSIM [12, 48] metrics. FVD and KVD employ the I3D [3] classifier pre-trained on the Kinetics-400 dataset [17]. For audio evaluation, we employ FAD [36] to gauge the distance between the features of the generated audio and the reference audio. We also use our proposed AVHScore to measure the alignment degree of the generated results. For all evaluations, we generate a random sample for each text without any automatic ranking.

## 5.2 Main results

**Comparison methods setting.** To the best of our knowledge, there are no existing usable methods directly relevant to our proposed task for comparison. Therefore, we combine existing related models and design two-stage methods for comparison.

(1) AnimateDiff [10]+AudioLDM [25]: Input text and utilize these two models to generate audio and video respectively.

(2) AnimateDiff [10]+Diff-Foley [26]: Input text, employ AnimateDiff to generate video, and subsequently utilize Diff-Foley to generate audio based on the video.

(3) AudioLDM [25]+TempoToken [54]: Input text, use AudioLDM to generate audio, and then employ TempoToken to generate video based on the audio.

**Quantitative comparison.** We present the quantitative results of our method alongside comparison methods for the TAVGBench and FAVDBench datasets in Table 2. The results demonstrate that our TAVDiffusion model outperforms all comparisons in terms of video and audio quality metrics. Specifically, the scores for FVD and KVD are 776.25 and 65.53, respectively, while the FAD score is 1.46. This demonstrates significant consistency between the audible video we generated and the original content and is of higher quality. These results underscore a significant consistency between the videos generated by our model and the original content, indicating superior quality. Additionally, our model achieves a notable CLIPSIM score (24.18), reinforcing the semantic coherence between the generated

**Table 2: Quantitative comparison. The numbers in the left column of the table represent the comparison methods we described in the previous section.**

| Methods | TAVGBench | | | | | FAVDBench [38] | | | | |
|---|---|---|---|---|---|---|---|---|---|---|
| | FVD ↓ | KVD ↓ | CLIPSIM ↑ | FAD ↓ | AVHScore ↑ | FVD ↓ | KVD ↓ | CLIPSIM ↑ | FAD ↓ | AVHScore ↑ |
| (1) | 777.54 | 70.04 | 22.49 | 1.62 | 10.17 | 867.58 | 115.81 | 22.11 | 1.67 | 12.13 |
| (2) | 777.54 | 70.04 | 22.49 | 1.81 | 12.28 | 867.58 | 115.81 | 22.11 | 1.89 | 15.07 |
| (3) | 1624.81 | 88.61 | 14.87 | 2.12 | 6.38 | 2640.96 | 591.43 | 15.16 | 2.29 | 6.12 |
| Ours | **516.56** | **45.76** | **25.44** | **1.38** | **23.35** | **776.25** | **104.26** | **24.18** | **1.46** | **29.06** |

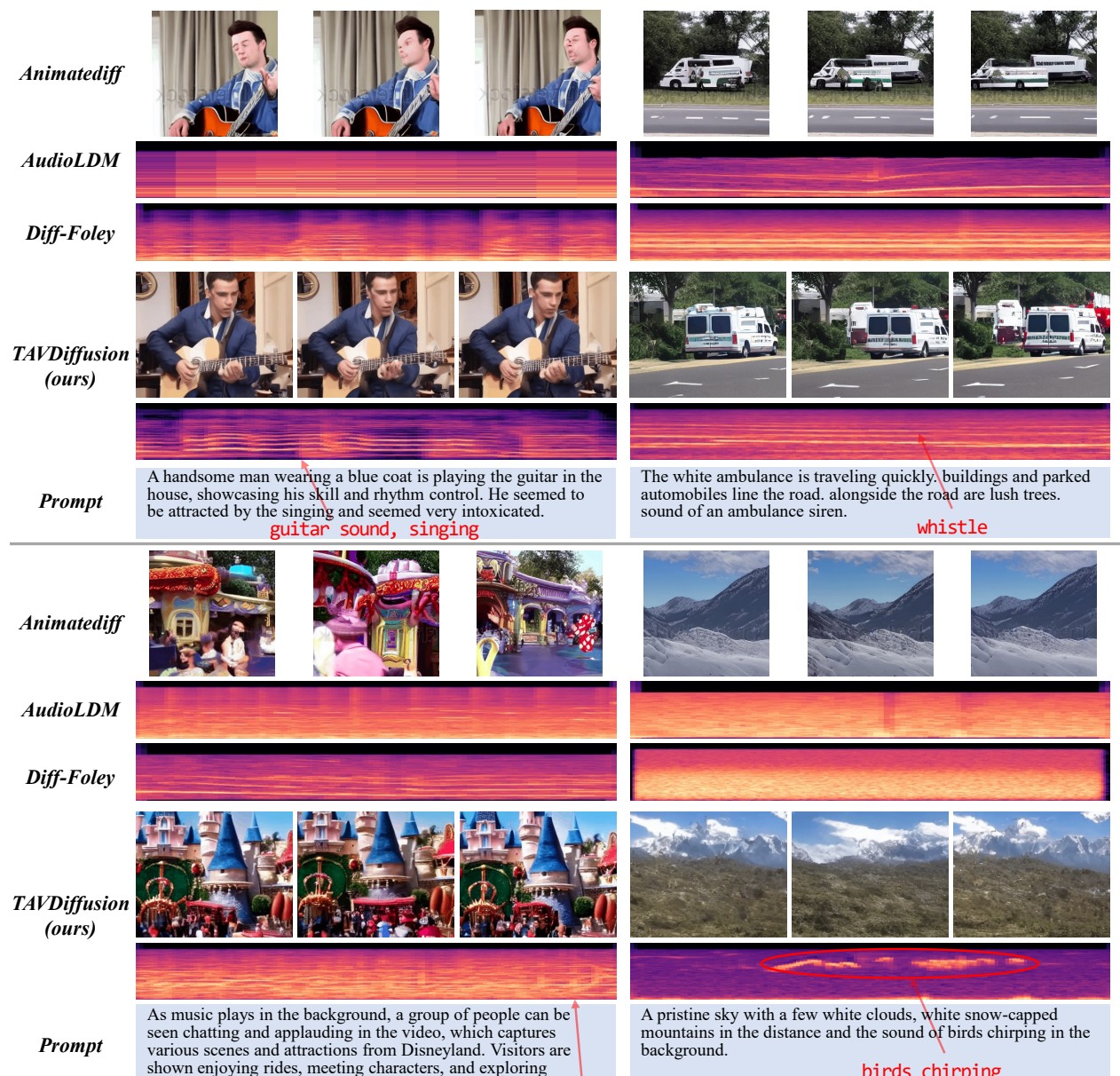

**Figure 5: Qualitative comparison. We compare our TAVDiffusion model with methods (1) and (2). Given the inferior visual quality of method (3) in our task (see Table 2), we exclude it from the qualitative comparison. Best viewed on screen.**

**Table 3: Ablation studies. "C.A." represents cross-attention mechanism, and $\mathcal{L}_{\text{EAS}}$ represents our proposed explicit audio-visual alignment strategy.**

| | | TAVGBench | | | | | FAVDBench [38] | | | | |
|---|---|---|---|---|---|---|---|---|---|---|---|
| C.A. | $\mathcal{L}_{\text{EAS}}$ | FVD ↓ | KVD ↓ | CLIPSIM ↑ | FAD ↓ | AVHScore ↑ | FVD ↓ | KVD ↓ | CLIPSIM ↑ | FAD ↓ | AVHScore ↑ |
| ✗ | ✗ | 687.23 | 66.21 | 23.02 | 1.58 | 14.59 | 843.19 | 112.07 | 22.58 | 1.62 | 14.67 |
| ✗ | ✓ | 614.28 | 59.23 | 23.89 | 1.46 | 18.69 | 819.54 | 109.21 | 22.64 | 1.54 | 19.76 |
| ✓ | ✗ | 589.71 | 51.44 | 24.68 | 1.51 | 19.88 | 791.57 | 106.95 | 23.98 | 1.51 | 20.59 |
| ✓ | ✓ | **516.56** | **45.76** | **25.44** | **1.38** | **23.35** | **776.25** | **104.26** | **24.18** | **1.46** | **29.06** |

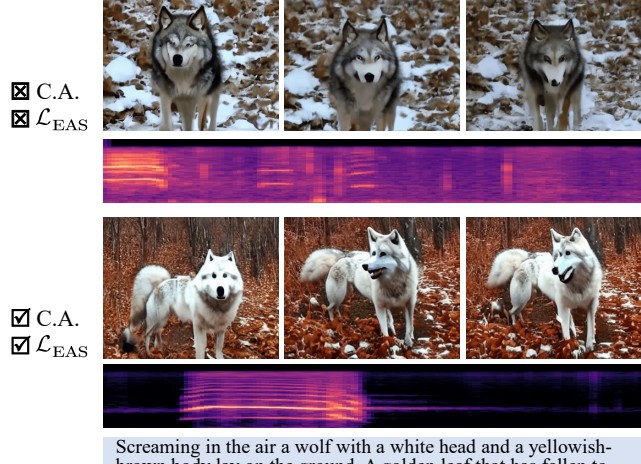

☒ C.A.
☒ $\mathcal{L}_{\text{EAS}}$

☑ C.A.
☑ $\mathcal{L}_{\text{EAS}}$

Prompt: Screaming in the air a wolf with a white head and a yellowish-brown body lay on the ground. A golden leaf that has fallen to the ground is covered in snow. The wolf kept barking and it was really loud.

**Figure 6: Qualitative comparison for the ablation studies. Best viewed on screen.**

videos and their associated prompts. Significantly, our model attains an AVHScore of 29.06, further evidencing its ability to generate videos with closely aligned audio and visual components. Note that neither our model nor the comparison models were exposed to the FAVDBench data during the training phase, thereby the results on this dataset further underscore our zero-shot capabilities.

**Qualitative comparison.** In Fig. 5, we present qualitative results comparing our method to the compared generators. The figure illustrates that TAVDiffusion outperforms the comparison models in terms of visual fidelity and the alignment of text, video, and audio. In the first example, the "performer" created by TAVDiffusion displays significantly enhanced realism, especially in facial expressions and hand movements. The generated audio also follows the two elements "guitar sound" and "singing voice" in the prompt. In the second example, TAVDiffusion showcases its ability to produce complex real-life scenes, maintaining the precise shapes of essential objects. It skillfully navigates the dynamics between the foreground object (*e.g.* the car) and the background scene, complemented by realistic audio. We also showcase our model's performance in two distinct scenarios: environments with significant background noise and notably quieter environments. For the former, our model generates various types of audio, such as music and human cheers, as directed by the prompt. For the latter, our model uniquely and accurately produces the sound of "birds chirping". This comparison shows the versatility of the model, demonstrating its effectiveness across a broad spectrum of audio-visual scenes. By evaluating the

model in such contrasting settings, we highlight its universal applicability and robustness in processing diverse auditory and visual inputs. We sincerely hope the readers find more examples of audible videos in the supplementary materials.

### 5.3 Ablation studies

To demonstrate the effectiveness of our proposed modules, we conduct ablation studies from both quantitative metrics (see Table 3) and qualitative visualizations (see Fig. 6). In Table 3, it is evident that the two strategies we proposed, multimodal cross-attention and multimodal alignment, enhance both the quality of video and audio generation and the alignment score. In Fig. 6, we can obverse that the "wolf" produced by our final model appears more realistic than that of the comparisons, with its mouth movements accurately reflecting the "kept barking" prompt and the generated audio. Please refer to the supplementary material for more samples.

### 5.4 Potential applications

Our TAVGBench and baseline model TAVDiffusion have a wide range of multimedia application fields. Our dataset, which includes large-scale video, audio, and corresponding text descriptions, is well-suited for a diverse range of multimodal tasks. It allows for the simultaneous use of text and audio as prompts to generate videos. Additionally, TAVGBench can be used to train audible video captioning models [38, 57] can substantially reduce the impact of insufficient audio-video-text data pairs as mentioned in [38].

## 6 CONCLUSION

We explored the challenge of creating videos with matching audio from text descriptions, a task known as Text to Audible-Video Generation (TAVG). To aid in this research, we introduced a new benchmark called TAVGBench, filled with over 1.7 million video clips. This resource is designed to help improve and evaluate TAVG models. We developed a method to automatically describe each audio-visual element, ensuring detailed and useful annotations for researchers. A new metric called Audio-Visual Harmoni score (AVHScore) was designed to evaluate the alignment of generated audible video. We introduced TAVDiffusion, a baseline model leveraging latent diffusion. This model incorporates cross-attention and contrastive learning mechanisms to achieve audio-visual alignment within the diffusion UNet framework. Extensive experimental results verified the effectiveness of our proposed framework, thus opening new avenues for multimedia content creation. In the future, we aim to explore a multimodal diffusion transformer [31] to facilitate audible video generation through a unified architecture.

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
