# OpenReview forum: "TAVGBench: Benchmarking Text to Audible-Video Generation"
_acmmm.org/ACMMM/2024/Conference — MM2024 Poster_

### Official Review · Reviewer_UZhg · 2024-05-19

**Rating:** 4
**Confidence:** 3

**Summary:**

The Text to Audible-Video Generation (TAVG) task entails creating videos with synchronized audio from text descriptions, which demands careful coordination of audio and video components. This paper develops a comprehensive Text to Audible-Video Generation Benchmark to support the Text to Audible-Video Generation task. This paper also proposes an automatic annotation pipeline to ensure each audible video has detailed descriptions
and introduces the Audio-Visual Harmoni score to provide a quantitative measure of the alignment between the generated audio and video modalities. Additionally, this paper presents a baseline model named TAVDiffusion for TAVG.

**Strengths:**

1.	This paper introduces a large-scale Text to Audible-Video Generation Benchmark, which contains over 1.7 million clips with a total duration of 11.8 thousand hours. It can advance the research of the Text to Audible-Video Generation task.
2.	This paper designs an automatic annotation pipeline to save costs for TAVG.
3.	This paper proposes a diffusion-based model and a quantitative metric for TAVG.

**Limitations:**

1.	The proposed automatic annotation pipeline is based on powerful open-source models. It is easy to follow. However, due to the large volume of video data, even automated annotation requires significant computational resources. Please provide clear information on the total annotation time and costs for the dataset in Section 3.2.
2.	Open-sourcing the entire dataset and related code would be more beneficial to the field.
3.	BLIP2 is capable of performing image captioning rather than video captioning tasks, so I'm curious how the paper utilizes BLIP2 to generate captions for a video.
I have two hypotheses. The first is that the features of multiple image frames are simply stacked.
In the original BLIP2,  a 224x224 resolution image correspond to one cls token and 256 patch tokens, which interact with 32 learnable tokens and image tokens. However, if tokens from different frames are merely stacked together, wouldn’t this approach neglect the frame-wise temporal modeling? Also, would using the default 32 tokens for videos result in information loss?
The second hypothesis is that a random frame is selected from a clip. However, for a clipped clip, the frames are still changing, and I wonder if randomly selecting one frame is reasonable?
Please provide additional explanation in section 3.2 Annotation details, and explain why a pretrained video caption model is not used.

**Suitability:**

3

---

### Official Review · Reviewer_j5hF · 2024-05-24

**Rating:** 3
**Confidence:** 3

**Summary:**

The authors have developed a comprehensive Text to Audible-Video Generation Benchmark (TAVGBench), which contains over 1.7 million clips with a total duration of 11.8 thousand hours. They propose an automatic annotation pipeline to ensure each audible video has detailed descriptions for both its audio and video contents. Additionally, they introduce the Audio-Visual Harmoni Score (AVHScore) to provide a quantitative measure of the alignment between the generated audio and video modalities. Furthermore, they present a baseline model for TAVGBench called TAVDiffusion, which uses a two-stream latent diffusion model as a fundamental starting point for further research in this area.

**Strengths:**

TAVGBench introduces a new benchmark for the Text to Audible-Video Generation (TAVG) task. This task is an innovative extension of traditional text-to-video generation by integrating synchronized audio components. This benchmark fills a crucial gap in current research, which predominantly focuses on unimodal video generation.
The dataset includes over 1.7 million clips with a total duration of 11.8 thousand hours, making it one of the largest datasets for this type of multimodal generation task. The size and diversity of the dataset ensure that models trained on it can generalize well to a wide range of real-world scenarios.
The paper is well-structured and clearly presents the problem, methodology, and results. The use of figures, such as the comparison of TAVG with existing tasks and the detailed annotation pipeline, helps in understanding the complexities and innovations introduced.

**Limitations:**

The automatic annotation pipeline, though efficient, may still result in inaccuracies or inconsistencies in the annotations. While using models like BLIP2, WavCaps, and ChatGPT improves quality, human oversight and correction might still be necessary to ensure the highest accuracy and relevance of annotations.
The paper introduces TAVDiffusion as a baseline model, which is a strong starting point. However, having only one baseline model may not provide a comprehensive understanding of the task. Including multiple baseline models or comparing with existing state-of-the-art models in related fields (e.g., text-to-video or text-to-speech) would provide a more robust evaluation framework.
There is a lack of comparison with the results of current advanced methods, and the visualization results lack complex scenes and action changes.

**Suitability:**

3

---

### Official Review · Reviewer_MB7j · 2024-05-24

**Rating:** 2
**Confidence:** 3

**Summary:**

The paper describes a benchmark dataset designed to generate videos and their corresponding audio from text prompts. The creation of the dataset is fully automated, with 1.7 million clips available. The author further trains a baseline diffusion model with audio-video cross-attention and contrastive learning, which can generate both streams. The author proposes a new evaluation metric to measure the alignment of audio and video and conducts initial experiments on this new benchmark.

**Strengths:**

1. The author introduces a relatively new task that involves generating a video and its corresponding audio using the same multimodal generation model. This is intriguing because future multimodal generation should be capable of producing both streams to ensure coherence and internal alignment.

2. The author then constructs a large-scale, automatically labeled dataset to facilitate the training of the model.

3. The author suggests a baseline multimodal diffusion model that begins with the same text prompt, generates both streams simultaneously, and uses a cross-attention alignment loss of the latent feature to aid in producing more coherent results. This could serve as a solid starting point model.

**Limitations:**

1. I have concerns about the quality of the data. Given that the process is fully automated, the data quality is determined by the audio caption and video caption models, both of which are separate and can introduce their own inaccuracies and hallucinations. The generated prompts from both models are then fed into ChatGPT to integrate them and correct grammatical errors, among other things. It's crucial to note that ChatGPT lacks multimodal capabilities, and the audio/video features are not connected to the LLM in any way other than the captions produced by the two separate models. At best, the LLM can reduce grammatical errors and enhance text readability, but it cannot address any hallucination issues arising from errors in the audio/video models. Thus the data should be treated as weak label/noisy data.

2. Furthermore, the author uses samples from the dataset as an evaluation benchmark, with no mention of any filtering or quality control. Regardless of the model's score on its test set, a model trained/evaluated on this dataset will likely overfit to the errors introduced by the caption models and cannot serve as a reliable benchmark.

3. Regarding the baseline model analysis, it's not clear of the settings of the baseline model. Are these model trained under the same constructed dataset, or just used in evaluation only? Since the training/eval dataset has the similar distribution, it's not fair to compare those model in the eval set.

**Suitability:**

3

---

### Official Review · Reviewer_uf3x · 2024-05-26

**Rating:** 5
**Confidence:** 3

**Summary:**

This paper focuses on the Text to Audible-Video Generation (TAVG) task, which involves creating videos with synchronized audio from textual descriptions. This task necessitates precise alignment of audio and video outputs to ensure a cohesive viewing experience. To facilitate research in this area, the authors have developed the Text to Audible-Video Generation Benchmark (TAVGBench), a substantial resource featuring over 1.7 million clips and 11,800 total hours of content. The authors also propose a baseline model —TAVDiffusion to verify the collected dataset.

**Strengths:**

This paper proposes a large-scale TAVG dataset with 1.7M Clips, which is valuable for future research T2AV generation.
The paper is written clearly, and the dataset description contains comprehensive details.

**Limitations:**

Lacking the quality analysis of automatic generated captions.
The discussion of potential applications is somewhat limited in scope, which raises doubts about the broader impact and applicability to downstream tasks.
The cases included in the paper contains a lot of artifacts which compromise the effectiveness of the proposed TAVDiffusion model.

**Suitability:**

2

---

### Meta-Review · Area_Chair_pEYE · 2024-07-03

**Recommendation:** Accept (Poster)
**Confidence:** 5

**Metareview:**

The authors have developed the Text to Audible-Video Generation Benchmark (TAVGBench), a substantial resource featuring over 1.7 million clips and 11,800 total hours of content, which is valuable for future research on Text to Audible-Video Generation.

However, the proposed annotation pipeline, even with the quality control discussed in the rebuttal, may result in inaccuracies in the annotations, which need to be discussed and/or further addressed. To verify the developed dataset, the authors proposed a model called TACDiffusion. Regarding the comparison with the SOTA, the comparison may be extended to include the latest video captioning models.